

# Benefits of km-scale climate modeling for winds in complex terrain: strong versus weak winds

Danijel Belušić[1,2], Petter Lind[1]

[1]Swedish Meteorological and Hydrological Institute (SMHI), Rossby Centre, Norrköping, 601 76, Sweden
[2]University of Zagreb, Faculty of Science, Department of Geophysics, Zagreb, 10000, Croatia

*Correspondence to*: Danijel Belušić (danijel.belusic@smhi.se)

**Abstract.** The existence of many different wind types in complex terrain and the difficulty of obtaining representative wind observations hinder the analysis of the general benefits of high-resolution climate modeling for winds. We show that the added value of km-scale modeling is particularly pronounced in mountainous terrain and increases substantially with wind speed, with the km-scale model and observations reaching twice larger speeds than a coarser model with 12 km grid spacing. At the same time, synoptically calm conditions are prone to local thermally generated circulations with typically weak winds, whose modeling results can also be considerably affected by the model resolution. We therefore focus on the mountainous region of the southern Scandinavian mountains and analyze the winds at two ends of the wind distribution: very strong winds, generally forced by large-scale weather systems, and local, thermally generated winds in synoptically calm conditions. Strong winds in the present climate are influenced more by the terrain height and high model resolution than by the large-scale forcing, while the future change is mostly governed by the global-model large-scale circulation change. For the thermal winds in summer, in contrast to the coarse model, the km-scale model captures glacier downslope wind in the high mountains and the resulting convergence zone as well as the increased cloud cover where the glacier wind meets the daytime upslope wind. The future change in thermal winds is primarily influenced by the future temperature change and the high model resolution. Since the future temperature changes are considerably less uncertain than the changes in large-scale circulation, the future of local weak thermal winds can be estimated with less uncertainty compared to stronger winds.

## 1 Introduction

Most studies dealing with km-scale, or convection-permitting, regional climate models (CPMs) have focused on precipitation and the associated mechanisms, especially in the present climate (e.g. Berthou et al., 2018; Ban et al., 2021). The considerable added value of CPMs compared to their parent regional climate models (RCMs) or global climate models (GCMs) is well established in the present climate for convective precipitation, especially for sub-daily heavy precipitation events (e.g., Prein et al., 2015; Lind et al., 2020; Ban et al., 2021; Lucas-Picher et al., 2021).
Studies on CPM winds, and especially their future changes, are less common. One reason for this is that winds generated or influenced by heterogeneous terrain often scale with the spatial dimensions of terrain features. In the complex terrain of the





Scandinavian mountains, which is the focus of this study, many spatial scales interact, as do the different types of terrain-induced circulations. The general difficulty of measuring the representative wind speed and direction for a certain area becomes even more apparent in such complex terrain. Therefore, evaluating the performance of km-scale regional climate models in reproducing wind is challenging, and climate modeling studies have addressed wind much less frequently compared to other variables such as temperature and precipitation. However, it has been shown that km-scale resolution is

required in complex terrain to adequately simulate observed wind patterns (e.g. Wang et al., 2013; Cholette et al., 2015; Belušić et al., 2018; Belušić-Vozila et al., 2023; Molina et al., 2024). Furthermore, enhanced convective clouds and precipitation have been linked to thermally-driven winds in complex terrain (e.g. Langhans et al., 2013; Cortés-Hernández et al., 2024).

Another reason for the relative scarcity of CPM wind studies is that the response of winds to climate change is highly

dependent on future projections of large-scale circulation patterns, which are particularly uncertain across the North Atlantic (e.g. IPCC, 2021; Little et al., 2023). However, some wind systems or wind characteristics are closely related to the terrain and are therefore sufficiently persistent to allow a dedicated study. Here we examine the added value of a km-scale regional climate model in simulating terrain-related winds in the present climate, as well as the potential added value (i.e. the difference to a coarser model) in simulating future change. In addition, we focus on two contrasting wind extremes: very

strong winds, which are typically large-scale winds accelerated by terrain; and very weak winds, which are typically thermal terrain-generated circulations in the form of nocturnal downslope (katabatic) and daytime upslope (anabatic) winds that occur in situations with weak synoptic flow. Both wind extremes can affect the local population and wildlife in different ways, e.g. through mechanical damage or increased fire risk during strong winds and through stagnant cold air pools or reduced air quality during weak winds.

**2 Data and methods**

**2.1 Model simulations**

The CPM simulations were carried out as part of the Nordic Convection Permitting Climate Projections (NorCP) project and are described in detail in Lind et al (2020, 2023). NorCP is based on the regional climate modeling system HARMONIE-Climate cycle 38 (HCLIM38; Belušić et al., 2020), which consists of two physical configurations: HCLIM38-ALADIN,

which is used as a regional climate model (RCM) with a horizontal grid spacing of 12 km, and HCLIM38-AROME, which is used as a CPM with a horizontal grid spacing of 3 km. HCLIM38-ALADIN is used as an intermediate model to bridge the gap in grid spacing between a global climate model (GCM) and the CPM.

The NorCP climate projections consist of 20-year time slices in three periods: historical (1986-2005), mid-century (2041-2060) and end-of-century (2081-2100), downscaling two CMIP5 GCMs: EC-Earth (Hazeleger et al., 2010, 2011) and

GFDL-CM3 (Griffies et al., 2011; Donner et al., 2011). The 21-year long evaluation simulation (1998-2018) downscaled ERA-Interim (Dee et al., 2011). All simulations were initiated one year before the mentioned time slices as the model spin-



up. We focus on the RCP8.5 emission scenario because it provides the largest signal-to-noise ratio, especially given the rather short time-slice simulations.

The analyzed model variables are from the near-surface or surface output, with wind speed given at the standard height of 10

m. However, the vertical velocity, which is only available in HCLIM3, is estimated from the model output at constant pressure levels, using the pressure level that is first above the local terrain height at each model grid point and at each output time.

## 2.2 Observations, reanalysis and evaluation

Two main sources of historical wind data are used to evaluate the HCLIM model: ERA5 reanalysis data (Hersbach et al.,

2020), and in-situ wind speed observations from a large number of meteorological observation stations in Sweden (SMHI, 2025). The wind speed observations from the SMHI observation network comprise a total of 161 stations. These observations are made at 10 m above the ground with a 3-hourly output frequency, and we have analyzed daily mean values. We use quantile-quantile (Q-Q) plots to compare the distributions of daily mean wind speed values between the two models, ERA5 and the station observations. Each point in a Q-Q plot shows the values of the same quantile from two different

datasets, with the final graph providing a clear representation of the relationship between the distributions of the two datasets. Datasets with two identical distributions would follow the identity (y=x) line indicated in the corresponding figures below.

## 2.3 Selection of strong- and weak-wind situations

Strong winds are selected using the 95th percentile of daily wind maxima.

To detect the thermally forced circulation, the analysis is limited to situations with weak synoptic gradients, i.e. weak large-scale winds. For each model grid point, the cases below the 5th percentile of hourly wind speed at 700 hPa are selected and averaged. The final results were tested for robustness to the choice of percentile, and the use of the 2nd percentile gave very similar results. The selection was made separately for each season and each climate period (historical, mid-century, end-of-century). To focus on typical conditions that lead to downslope and upslope winds, we use two representative situations: DJF

at 00 UTC for stable conditions and JJA at 12 UTC for unstable conditions, respectively.

A subdomain covering the southern Scandinavian Mountains is used for the analysis of strong and weak winds, as this is the region with the highest and most complex terrain and this choice limits the otherwise large latitudinal change of the entire Scandinavian Mountains (about 10 degrees of latitude with a length of about 1700 km).



## 3 Results

### 3.1 Evaluation of simulated wind speed

The evaluation of the model results in relation to the station observations and ERA5 is carried out separately for inland stations with predominantly flat terrain, coastal stations and mountain stations (Fig. 1). Both HCLIM12 and HCLIM3 have similar frequency distributions as the station observations for inland stations, and there is no clear added value in the km-scale simulation. Interestingly, ERA5 somewhat overestimates wind speeds for all but the strongest winds in winter. For coastal stations, the models and ERA5 underestimate the strong winds, especially in winter, and the models somewhat underestimate the weak to moderate winds in both seasons, but the overall agreement with the observations is good.

For mountain stations, HCLIM12 and ERA5 considerably underestimate the moderate and strong winds, with the underestimation increasing with wind speed and reaching almost twice smaller values for strongest winds in winter for ERA5. On the other hand, HCLIM3 agrees very well with the observations, especially in winter. In summer, the wind speeds are slightly underestimated, which is more pronounced for very weak winds. These results clearly show that the km-scale resolution is necessary for modeling strong winds in complex terrain, and at the same time implies that the km-scale resolution could be sufficiently high to capture strong mountain winds.

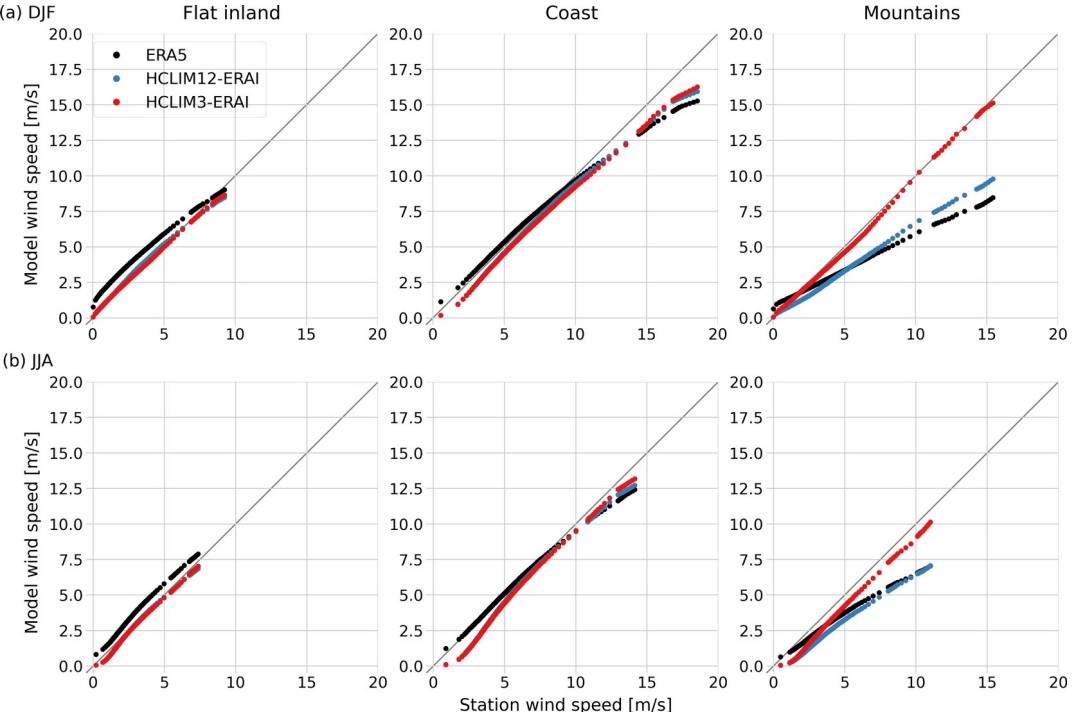

**Figure 1: Quantile-quantile plot of daily mean 10-m wind speed in SMHI station data (x-axis) vs. ERA5 reanalysis and HCLIM evaluation simulations (y-axis). Shown are DJF (top panels) and JJA (bottom panels), for inland stations (left panels), coastal stations (middle panels) and mountain stations (right panels). The identity (y=x) line is depicted in gray.**



### 3.2 Assessment of daily mean winds

The daily mean wind in the historical period is predominantly influenced by large-scale forcing, with the winds being stronger in winter (Fig. 2) than in summer (Fig. 3). The effect of the km-scale resolution is evident in the mountains, where the winds are generally stronger in HCLIM3.

The future change is predominantly governed by the forcing GCM in both seasons. HCLIM simulations forced by EC-Earth (HCLIM12-ECE, HCLIM3-ECE) have smaller changes compared to the GFDL-forced simulations (HCLIM12-GFDL,

HCLIM3-GFDL), which show a considerable wind strengthening in the southern part of the domain in winter and a general wind weakening in summer. The differences between HCLIM12 and HCLIM3 downscaling a single GCM are small compared to the differences between HCLIM12 or HCLIM3 forced with different GCMs.

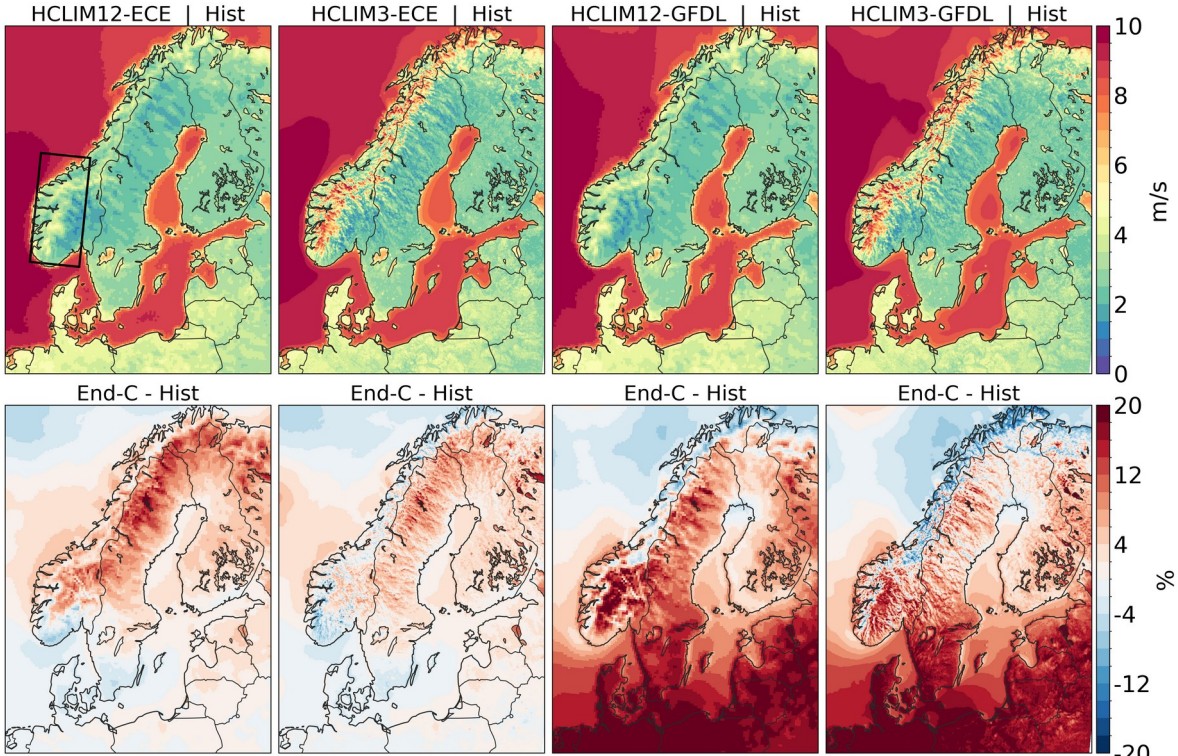

**Figure 2: DJF daily mean 10-m wind speed (top panels) and its percentage change by end-of-century (bottom panels) in HCLIM12 and HCLIM3, forced with EC-Earth and GFDL. The black rectangle over southern Norway depicts the domain used for investigating winds over complex terrain.**





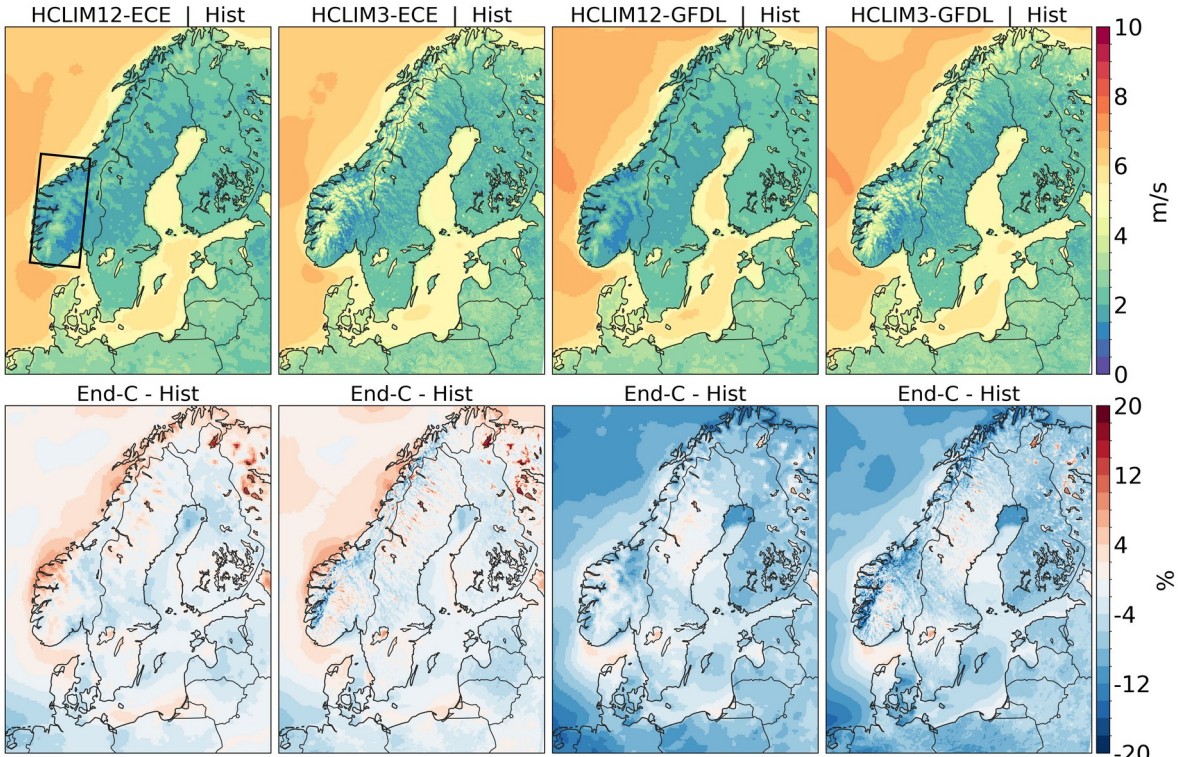

**Figure 3: As Fig. 2, but for JJA.**

### 3.3 Strong winds

Focusing on the strong winds, Fig. 4 shows that the HCLIM12 10-m wind speed gradually increases with terrain height and reaches the same values regardless of the forcing GCM, in both winter and summer. HCLIM3 is also independent of the GCM forcing, but the increase in wind speed with terrain height is much greater than for HCLIM12. For example, at a terrain height of 1500 m, the HCLIM3 wind speed reaches more than 20 m/s in winter and 14 m/s in summer, while HCLIM12 reaches only 10 m/s in winter and 7 m/s in summer, i.e. twice smaller values. This is consistent with the evaluation results for mountains in Fig. 1, but is even more pronounced here because the focus is on strong wind events.

However, the future change of strong winds is much more affected by the GCM forcing. This is particularly evident for the GFDL forcing in winter, where HCLIM12 and HCLIM3 show very similar future changes at all terrain heights, with wind speed increasing at all levels. In general, it appears that the magnitude of the future changes in both HCLIM12 and HCLIM3 is largely determined by the GCM forcing. However, there are notable differences between the two model resolutions in the dependence on terrain height of the wind speed changes that is likely dependent on model structure and/or grid resolutions. This is particularly evident in summer, where the future change is a general decrease in wind speed at all levels. The





percentage decrease is slightly larger for the GFDL forcing compared to the EC-Earth forcing. However, the magnitude of the relative decrease in HCLIM3 becomes consistently larger with terrain height, for both GFDL and EC-Earth forcing, while in HCLIM12 the magnitude is quasi-constant with terrain height (EC-Earth forcing) or even increasing at higher altitudes (GFDL forcing).

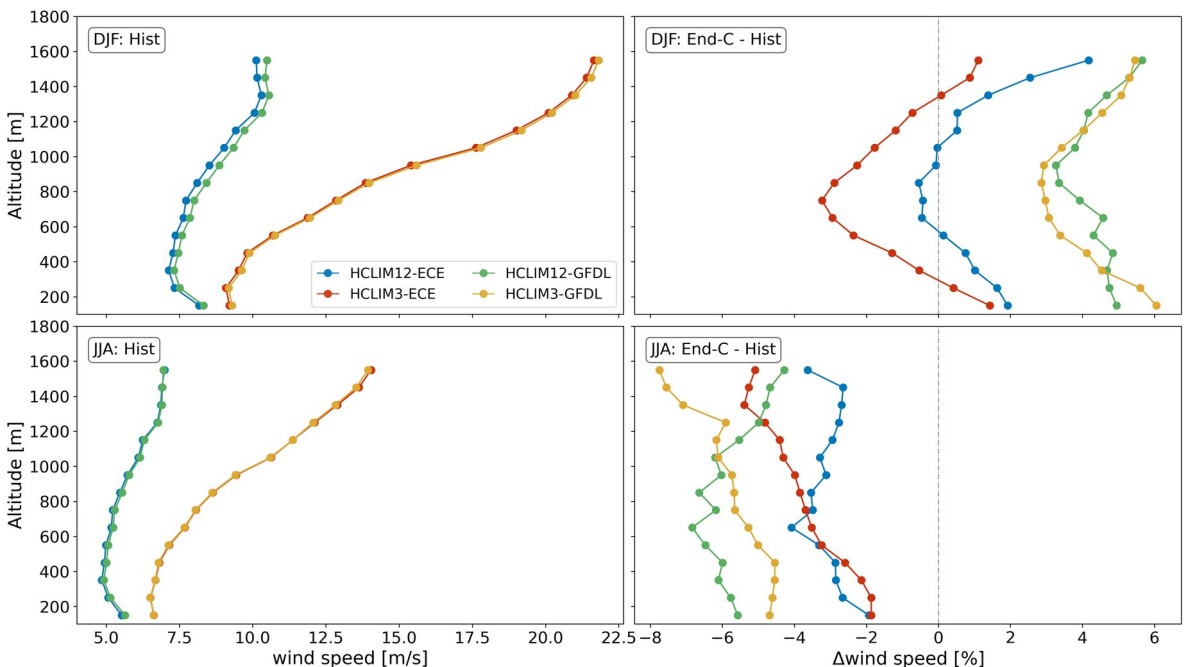


**Figure 4: The 95th percentile of daily maximum 10-m wind speed as a function of terrain height over the southern Norway subdomain (indicated in Fig. 2) in HCLIM3 and HCLIM12 scenarios for DJF (top panels) and JJA (bottom panels). Shown are the wind speeds in the historical period (left panels) and the change, in percent, by end-of-century (right panels). The data are bin-averaged over 15 100-m high vertical bins.**


### 3.4 Thermal circulations in large-scale weak-wind situations

   In large-scale weak wind situations, the thermal circulation systems in the mountains are generated by the terrain slope and buoyancy, with the latter typically expressed as the difference between the air and surface temperatures (e.g. Bintanja et al., 2014). In a nighttime stable boundary layer, the surface is colder than the air next to it due to radiative cooling, resulting in a

surface temperature deficit and negative buoyancy, which leads to katabatic (downslope) wind. In a daytime unstable boundary layer, the surface absorbs the incoming solar radiation and is warmer than the air, resulting in positive buoyancy and anabatic (upslope) wind. The presence of snow cover generally lowers the surface temperature compared to the equivalent conditions without snow cover and can cause downslope glacier wind even in daytime conditions. As typical



representatives of unstable and stable conditions, we choose summer daytime and winter nighttime situations, respectively,
for further analysis.

### 3.4.1 Unstable conditions

For summer daytime conditions, both HCLIM12 and HCLIM3 in the historical period have similar wind speeds for
mountain heights below about 1200 m, gradually decreasing with height (Fig. 5). Above 1200 m, the HCLIM12 wind speed
continues to decrease, while the HCLIM3 wind speed rather abruptly starts increasing with height.

The abrupt shift at about 1200 m in HCLIM3 can also be seen in the signal for the future changes. For lower terrain heights,
the future change in wind speed is consistently positive in both HCLIM3 and HCLIM12, except for GFDL-forced HCLIM12
below 300 m. For terrain heights above 1200 m, the HCLIM3 future change signal abruptly switches to negative values,
while HCLIM12 remains positive. This discrepancy between the RCM and the CPM is examined below.

Note that the exact value of the terrain height at which the change occurs, 1200 m, is determined approximately and varies
slightly between the different plots, e.g. it is about 100 m higher for the EC-Earth forcing than for the GFDL forcing.
However, as these are composite plots over 20-year periods and many grid points, exact correspondence in the dynamical
sense is not to be expected.

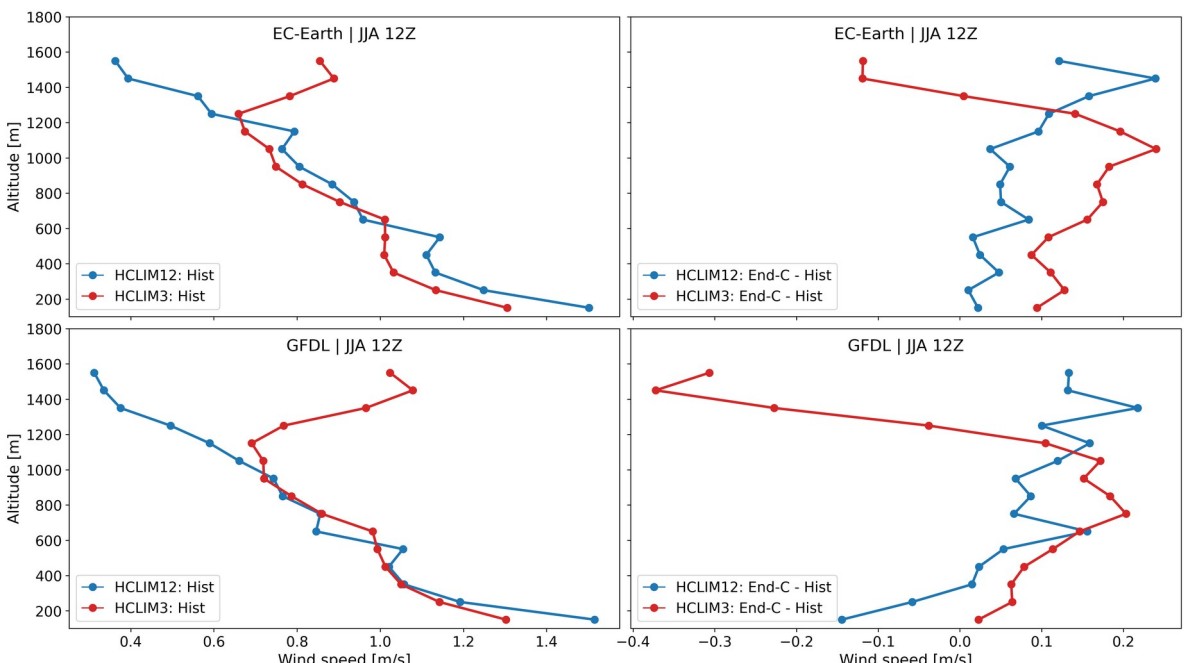

**Figure 5: HCLIM12 and HCLIM3 simulated JJA 12 UTC wind speed as a function of terrain height over the southern Norway
subdomain (indicated in Fig. 2) for large-scale weak-wind conditions (see text for explanation). Shown are the historical period
(left panels) and the changes by end-of-century (right panels) in HCLIM3-ECE (top panels) and HCLIM3-GFDL (bottom panels).
The data are bin-averaged over 15 100-m high vertical bins.**





The HCLIM3 vertical velocity is positive for terrain heights below 1200 m, confirming the upslope anabatic nature of the
summer daytime circulation (Fig. 6). However, the direction of the flow turns to downslope for higher terrain. The direction
change is clearly related to the change in snow cover, with the direction changing to downslope when the mean snow cover
increases to about 30% (Fig. 6). This suggests that the downslope flow direction is the result of the katabatic glacier wind,
since the presence of snow cover changes the sign of the surface temperature deficit, which is the main forcing for thermal
circulations on slopes (e.g. Grisogono and Oerlemans, 2001). This is seen from the vertical profiles of the temperature deficit
(Fig. 7), which has nearly constant negative values (positive buoyancy) below 1200 m, above which it rapidly increases and
crosses zero between 1400 and 1600 m, depending on the forcing GCM (the zero-crossing for the EC-Earth forcing is not
seen in the figure because the vertical axis is clipped below 1600 m due to too few grid points in HCLIM12 aloft).
HCLIM12 has a similar vertical profile of the temperature deficit, but the increase above 1200 m is weaker.

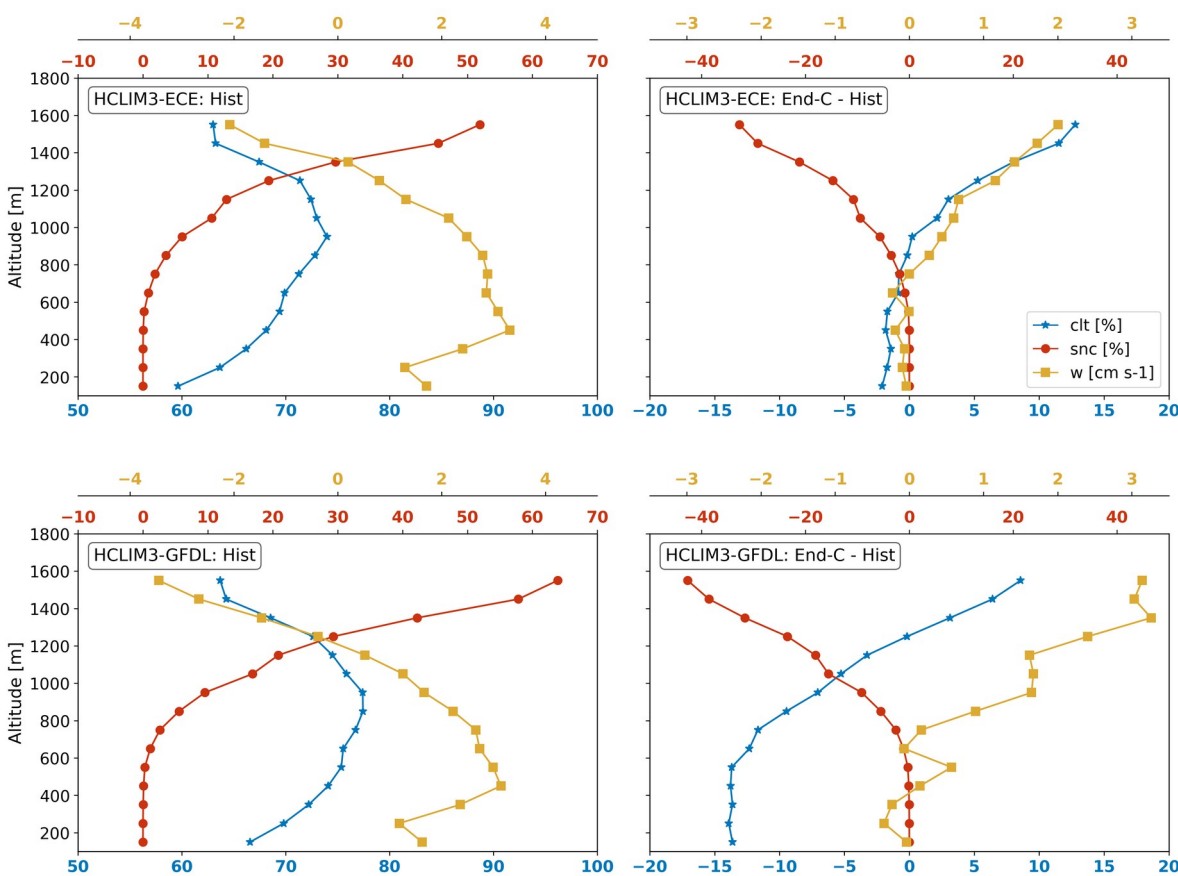


**Figure 6: HCLIM3 simulated JJA 12 UTC cloud cover (blue line/stars), snow cover (red line/circles) and vertical wind speed
(yellow line/squares) as a function of terrain height over the southern Norway subdomain (indicated in Fig. 2) for large-scale
weak-wind conditions (see text for explanation). Shown are the historical period (left panels) and the changes by the end of century
in RCP8.5 (right panels) in HCLIM3-ECE (top panels) and HCLIM3-GFDL (bottom panels). The data are bin-averaged over 15
100-m high vertical bins.**



Vertical velocity is not available for HCLIM12, but horizontal wind divergence can be used to detect differences in flow regimes between HCLIM12 and HCLIM3. The horizontal wind divergence in HCLIM3 (Fig. 8), which indicates a developed downslope flow over at least two grid points on a slope, occurs above 1400 m, where the snow cover exceeds

40%. This has important implications for understanding the differences between the RCM and the CPM. Both HCLIM12 and HCLIM3 have grid points with snow cover above 40%. However, analysis of the spatial patterns (not shown) reveals that HCLIM12, unlike HCLIM3, predominantly has fewer than two such neighboring grid points on a single slope, which seems to prevent the development of divergence and consequently katabatic flow even at high terrain altitudes, i.e. near mountain tops. This also prevents HCLIM12 from reaching a bin-averaged positive temperature deficit in high terrain.

Consequently, HCLIM12 has an upslope flow at all levels that is gradually weakening with height (Fig. 5), i.e. there is weak convergence at all levels (Fig. 8). In HCLIM3, the upslope flow below 1200 m meets the downslope glacier flow from above and a stronger convergence zone develops, reaching its maximum at terrain heights between 1000 and 1200 m (Fig. 8).

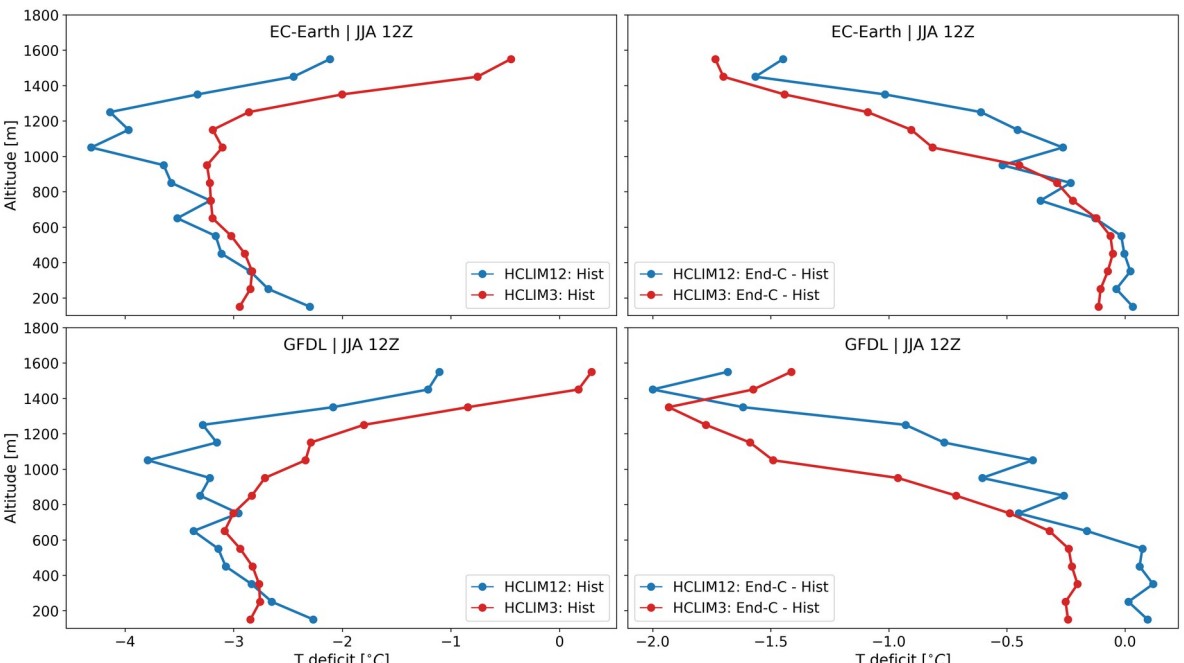

**Figure 7: As Fig. 5, but showing surface temperature deficit. The surface temperature deficit is defined as the difference between the temperature at the lowest model level (approximately 12 m above the terrain) and the surface radiation temperature.**

The existence of the convergence zone in HCLIM3 has an impact on cloud cover. HCLIM3 cloud cover gradually increases with height, reaching the maximum of about 80% at about 1000 m, exactly at the height of maximum convergence, and



decreases in the divergence regions above (Fig. 6). HCLIM12 has a gradual increase in cloud cover with height, without a
clear maximum and in line with the convergence at all heights (not shown).

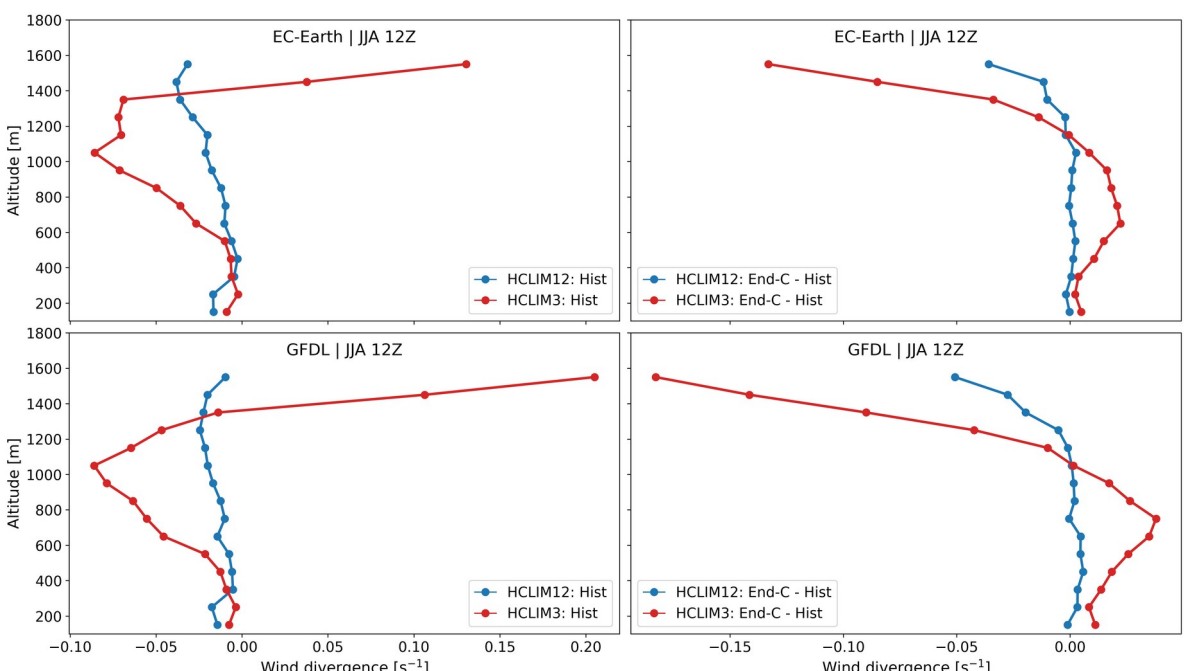

**Figure 8: As Fig. 5, but showing divergence.**


With future warming, the HCLIM3 temperature deficit decreases, indicating further destabilization of the surface layer (Fig.
7). The destabilization generally increases with terrain height up to about 1300 m. HCLIM12 has a similar future change
profile, except that there is weak stabilization for the lower terrain heights, especially for the GFDL forcing. The future wind
speed change is generally positive and increases approximately linearly with height (Fig. 5), which is consistent with the

increase of destabilization with height. There are two exceptions to the linear wind speed increase with height: the wind
speed decrease in the GFDL-forced HCLIM12 below 300 m, which is consistent with the low-level stabilization, and the
abrupt change in HCLIM3 occurring between 1000 and 1200 m, resulting in wind speed decrease aloft. For the latter, the
decrease in summer snow cover above 1200 m (Fig. 6) reduces the spatial extent and strength of the downslope flow in
HCLIM3. This is evident from the combined positive change in vertical velocity (Fig. 6) and the decrease in wind speed

(Fig. 5), which indicates the weakening of the downslope flow and its transition to higher terrain, together with the formation
of a weaker upslope flow below. As the divergence zone also weakens and moves to higher terrain (Fig. 8), the HCLIM3
cloud cover increases above 1200 m (Fig. 6). The changes in cloud cover in HCLIM12 are much smaller, despite the
comparable snowmelt (not shown). This is a further indication that the dynamics of the glacier wind was not reproduced by



HCLIM12 in the present climate, so that the future warming affects the mountain circulation much less. Namely, the future

warming and snowmelt in HCLIM12 result in the upslope wind strengthening at all levels, but without changing the nature or direction of the flow.

### 3.4.2 Stable conditions

The winter nighttime situation is simpler. The vertical velocity is negative at all terrain heights (not shown), so the flow is downslope as expected in stable conditions. There is a difference between the RCM and the CPM: the wind speed gradually

decreases with terrain height in the RCM, while it increases quite strongly in the CPM (Fig. 9). The latter is partly caused by much stronger winds in a few grid points in the CPM with very steep terrain or at the edges of glaciers, the proportion of which increases with height. In other words, this strong increase does not show up when the median is used instead of the mean to calculate the vertical bins, at least below 1000 m. This behavior is almost the same regardless of the forcing GCM. With future warming and the associated snowmelt, the surface is destabilized, leading to weaker winds at all heights. The

relative wind weakening is similar at all terrain heights, ranging between 10 and 20%.

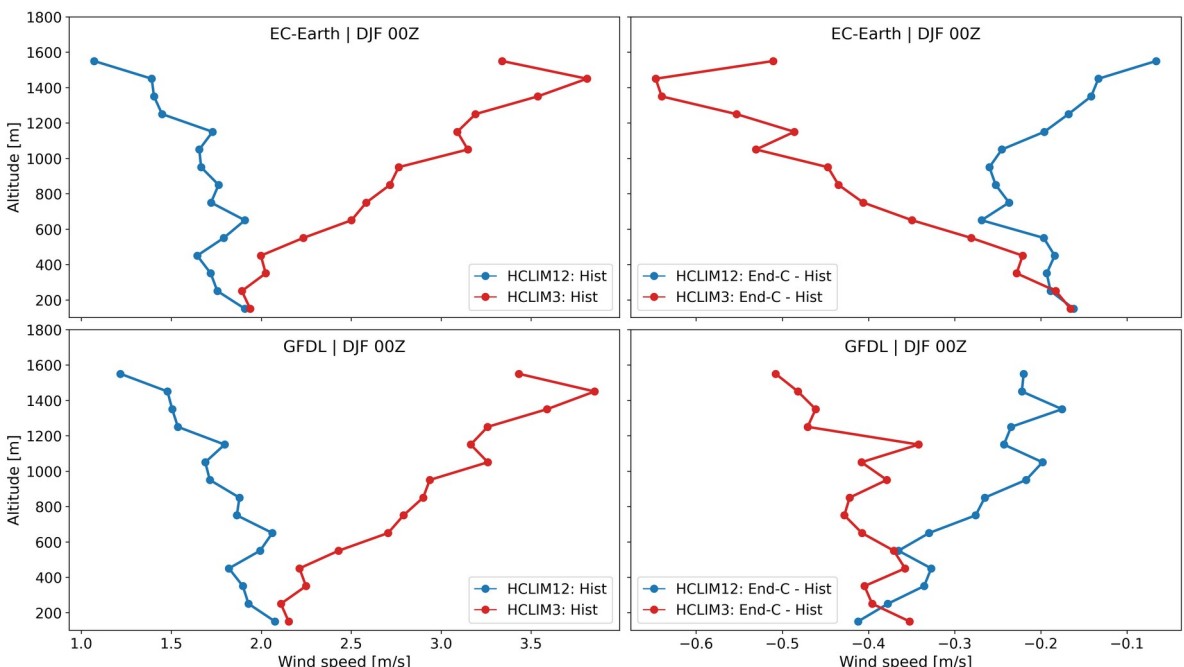

**Figure 9: As Fig. 5, but for DJF at 00 UTC.**



## 4 Conclusions

Benefits of km-scale modeling for winds depend on the complexity of the terrain, the nature of wind and the type of signal that is addressed, the latter being either the behavior in a given climate period or the change between different climates. The expected general conclusion is that the greatest added value of higher resolution modeling is found in complex mountainous terrain, which is a direct consequence of the better representation of the terrain. However, the evaluation also shows that the km-scale resolution might be sufficiently high to capture strong winds in complex terrain, at least when considering the setup of the available observation stations. The latter is of course highly dependent on the local terrain, and in many locations with complex terrain at small scales, sub-km resolution is advantageous or even crucial (e.g. Wang et al., 2013).

All wind types show benefits of km-scale modeling in complex terrain, but in different ways. For the mean and strong winds, the differences in wind speed can be very large, especially in higher terrain, reaching twice stronger winds in the km-scale model than in the RCM. In the case of weak, thermally generated winds, the km-scale model can reproduce the downslope glacier wind and the resulting divergence in high terrain, and the convergence in and around the frontal zone where the downslope glacier and upslope daytime winds meet in unstable summer daytime conditions. This affects the vertical distribution of cloud cover, which reaches its maximum in the convergence zone, which is consistent with the glacier fronts in the Himalayas (Lin et al., 2021). The RCM fails to reproduce the glacier wind. In stable winter nighttime conditions, both models reproduce the downslope katabatic flow, albeit with different terrain dependence: the strengthening of the wind with terrain height is only evident in the CPM.

The future change signal for the mean and strong winds is mostly influenced by the GCM forcing, in that the GCM forcing determines the magnitude of the wind change, while the vertical dependence of the change signal is resolution-dependent. For weak, thermally generated winds under unstable conditions, future warming with melting glaciers affects the RCM and CPM differently. Since the km-scale model reproduces the glacier wind, the future warming and the associated snowmelt cause the glacier wind to cease or decrease, weakening the convergence zone and shifting it to higher terrain. On the other hand, since there is no downslope glacier wind in the RCM in the historical period, the snow melt in the RCM leads to a strengthening of the already existing upslope flow. Under stable conditions, the future wind weakening is a consequence of destabilization due to warming and snowmelt and is similar in both models. Both the general strengthening of the daytime upslope wind and the weakening of the nighttime downslope wind with future warming are consistent with the results obtained for the Rocky Mountains in the USA (Letcher and Minder, 2017).

For thermal winds, the future change is only weakly dependent on GCM forcing, primarily due to the different GCM warming levels rather than large-scale circulation changes. Since future surface warming is a much less uncertain result compared to circulation changes, it may be possible to determine the fate of local thermal circulations in the future climate with comparatively high confidence.



**Code and data availability**

Data from meteorological observation stations in Sweden are available at: https://www.smhi.se/data/utforskaren-oppna-data/se-acmf-meteorologiska-observationer-vindhastighet-timvarde (SMHI, 2025). ERA5 data are available from the Copernicus Climate Data Store (CDS) on https://doi.org/10.24381/cds.adbb2d47 (Hersbach et al., 2023). Model outputs from the NorCP project are planned to be uploaded to ESGF. Currently the data are available upon request. Processed climate model and observational data, as well as the codes used for statistical calculations and visualization of data are archived in Zenodo at https://doi.org/10.5281/zenodo.15000594 (Lind and Belušić, 2025).

**Author contribution**

DB: conceptualization, formal analysis, methodology, writing – original draft preparation. PL: data curation, formal analysis, methodology, writing – review & editing.

**Competing interests**

The authors declare that they have no conflict of interest.

**Acknowledgment**

The regional climate simulations were performed within the NorCP project, which involved the Danish Meteorological Institute (DMI), Finnish Meteorological Institute (FMI), Norwegian meteorological institute (MET Norway) and the Swedish Meteorological and Hydrological Institute (SMHI). The authors acknowledge the use of computing and archive facilities at ECMWF and at the National Supercomputer Centre in Sweden (NSC) which is funded by the Swedish Research Council via Swedish National Infrastructure for Computing (SNIC).

**Financial support**

This work was partially supported by the Croatian Science Foundation under the project number HRZZ-IP-2022-10-9139, the Horizon 2020 project EUCP under Grant agreement no. 776613, and the Horizon Europe project ASPECT under Grant agreement no. 101081460.

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
