# Peer review of "Benefits of km-scale climate modeling for winds in complex terrain: strong versus weak winds"

_EGUsphere, 2025_

## Author Response (AR1)

We thank the two anonymous referees for carefully reading the manuscript and providing useful comments and suggestions. Below we respond to their comments point by point.

**Referee 1**

**- Related to the wind observations used, a more detailed description of the observational data used for model validation is needed. Please clarify:**
- **The number of stations used in each geographical category (inland flat areas, mountains, coastal zones), to ensure that all regions are equally represented.**
- **The spatial distribution of these stations, especially in relation to the domain's topography. The manuscript would be benefit of a map with the spatial distribution of the stations and its altitude.**
- **Any data filtering or quality control procedures applied in the treatment of outliers or continuity of time series.**

Response: We have now added Figure A1 in Appendix A of the manuscript which presents the locations of the SMHI stations in Sweden. The map also includes the orography height from the high-resolution model (HCLIM3). Part of Section 2.2 in the manuscript has been rewritten into the following:

"The wind speed observations from the SMHI observation network are made at 10 m above the ground with a 3-h output frequency, and we have analyzed daily mean values. All station data has undergone quality control and is used in SMHI's operational activities. Stations with more than 30% missing 3-h time steps have not been considered, leading to a total of 161 stations being used. Figure A1 in Appendix A shows the geographical locations of the stations. Based on their locations, e.g. orography height, these have been categorized into *inland flat, coastal* and *mountain* areas. Of the 161 stations, 108 were categorized as *inland flat*, 32 as *coastal* and 21 as *mountain* stations."

**- In the statement "Strong winds are selected using the 95th percentile of daily wind maxima," it is unclear which time period is used to define this percentile. Is this based on the historical simulation only, or the entire simulation ensemble? Please specify the temporal reference used to compute the threshold.**

Response: The percentiles are calculated for each period separately. We have clarified this in Section 2.3 accordingly:

"Strong winds are represented by the 95th percentile of the daily wind maxima, calculated for each time period (i.e. historical and end-of-century) and each grid point separately."

**- The description of the results should be more detailed, including numbers for the wind values in sentences such as: 'Interestingly, ERA5 somewhat overestimates wind speeds for all but the strongest winds in winter. What is considered moderate or strong wind in this case?**

Response: We agree that we should be more specific when talking about strong or weak wind speeds. Thus, we have added a definition of weak, moderate and strong winds for the daily mean wind speed analysis at the end of Section 2.2:

"In the analysis of daily mean wind speeds, we define "weak", "moderate" and "strong" wind speeds as < 5 m/s, 5-10 m/s and > 10 m/s respectively."

**- Since there is no direct measurement of the improvement of higher resolution, interpretation of results and comparison between simulations in sentences such as 'HCLIM3 is also independent of GCM forcing, but the increase in wind speed with terrain height is much larger than for HCLIM12' would benefit from a trend analysis such as the Mann-Kendal test.**
Response: We are not completely sure if we understood the referee's comment well. The analysis in Figure 1 provides a measure of added value in the high-resolution model. HCLIM12 and HCLIM3 perform similarly over flat inland and coastal areas which is to a large extent expected. However, the two models clearly deviate in performance over mountainous locations, where HCLIM3 shows much better agreement with observations. Regarding the cited sentence: it addresses how the wind speed varies with altitude in the control (historical) climate only, so it is not related to any changes between time periods. Hence, a Mann-Kendall test would typically not be applied here. Perhaps the manuscript text was unclear so now we emphasize that this discussion is related to the historical period. Otherwise, the larger wind speeds in HCLIM3, which become ever more pronounced at higher terrain altitudes, are clearly evident from Figure 4 (left panels).

**- The reasons of choosing the stable/unstable conditions in the next sentence 'The presence of snow cover generally lowers the surface temperature compared to the equivalent conditions without snow cover and can cause downslope glacier wind even in daytime conditions. As typical representatives of unstable and stable conditions, we choose summer daytime and winter nighttime situations, respectively, for further analysis.' could be better referenced.**
Response: We now provide references for these choices (namely, Van den Broeke, 1997; Zardi and Whiteman, 2013), together with the explanation that some model variables were output only every 6 h, thus additionally limiting the choice of hours. Since the first mention of this selection is in Section 2, this is where we provide the reference.

**- I miss a brief discussion of how ERA5 wind simulations compare to observations in previous studies, particularly in complex terrain. Are the known limitations of ERA5 reflected in your results? Referencing past evaluations (e.g., Olauson, 2018; Molina et al., 2021, Gutierrez et al., 2024) could help contextualize your findings.**
Response: As suggested by the reviewer, we have now included a brief discussion in Section 2.2 in the manuscript about the assessed performance of ERA5 for near-surface wind speeds, as follows:
"Several studies have assessed the near-surface winds in ERA5 on multiple time scales. These indicate that, in general, ERA5 is able to capture the frequency distributions and average wind speeds, from hourly to seasonal time scales (e.g. Chen et al., 2024; Molina et al., 2021; Fan et al., 2021). However, it has also been shown that ERA5 struggles with representing wind extremes, often overestimating weak winds and underestimating strong winds, and most

evidently in areas of complex terrain and in coastal regions (e.g. Belušić Vozila et al., 2024; Potisomporn et al., 2023; Gandoin and Garza, 2021)."
We also thank the reviewer for the suggested references.

**- The mentioned Figures not shown on the manuscript might be shown and referenced on the supplementary material.**
Response: The previously not shown figures are now included in Appendix A.

**Minor comments**

**- L60: Rewrite this for clarity: 'The 21-year long evaluation simulation (1998-2018) downscaled ERA-Interim (Dee et al., 2011)' to 'the 21-year long-term evaluation simulation is (1998-2018) reduced from ERA-Interim (Dee et al., 2011)'.**
Response: We thank the reviewer for this suggestion. To clarify this sentence, it was rewritten to: "The 21-year long evaluation simulation (1998-2018) uses ERA-Interim (Dee et al., 2011) as initial and boundary data."

**- L98: correct 'for the strongest winds'**
Response: Done

**- L140: 'percentage decrease' by percentage of decrease.**
Response: We have decided to keep the original formulation, because the figure depicts the change in percent, and therefore we refer to it as the percentage change (i.e. percentage decrease or decrease in percentage). "Percentage of decrease" would have a different meaning.

**- L202: correct to : 'neighbouring'.**
Response: Done

**- L233: correct to: 'the dynamics of the glacier wind were not reproduced'.**
Response: Done

**_Referee 2_**

**This manuscript explores the advantages of km-scale climate models for the simulation of wind patterns over Scandinavia. The authors prove evidence that km-scale models (dx=3km) are able to simulate the wind speeds over complex, mountainous terrain with a higher skill compared to models with dx=12km. The authors attribute this improvement to better-resolved topography and a more realistic simulation of thermally-induced circulations, which are common over complex terrain. The manuscript is generally well written and provides a valuable contribution to the (large) research question on which processes are relevant for the successful simulation of wind speeds over complex**

**topography. However, I have a few questions on the categorization and interpretation of the results, among some minor remarks.**

**Major comments**

**1) Interpretation of the thermally-induced circulations over mountainous terrain:**

**The authors argue that the 3km simulations have an improved representation of the thermally-induced circulation, and especially anabatic/katabatic slope winds. However, I would argue that the thermally-induced circulation (or Alpine pumping), also includes the large-scale plain-to-mountain circulation, and as a result, either up-valley (daytime) or down-valley (nighttime) flows (Zardi and Whiteman, 2013) are equally important in this study. I was surprised that these two circulations are not mentioned in the manuscript, because they even act on larger scales (ie., plain-to-mountain scales or entire valleys) than small-scale slope flows. This might have implications for the interpretation of the model results. Previous studies (Langhans et al, 2013; Graf et al, 2015) argue that thermally-induced flows are well-represented in km-scale climate models, in agreement with the findings of this manuscript.**

Response: Mountain-plain and, even more importantly for the southern Scandinavian mountains, mountain-ocean circulation are well reproduced in both resolutions, so large benefits are not expected from km-scale, which is consistent with Langhans et al. (2013). On the other hand, given that valley floor widths rarely surpass a few km, valley flows cannot be reproduced by the 12-km model and only in case of very wide valleys can be partially reproduced by the 3-km model. Therefore, with coarsening resolution valleys disappear, but slopes remain as long as a mountain remains (although of course smoothened due to low resolution). So slope flows tend to exist even when valley flows are absent. Our analysis explores the differences between the 12 and 3 km models on climate time scales from a general perspective, which in our case is chosen as a comparison over the same bin-averaged height intervals, and hence the focus is on slope flows. But we agree with the reviewer that other thermal mountain circulations could affect the results and discuss this in the paper now (Section 3.4).

**My second question would be the following: Since purely thermally-induced valley winds could reach wind speeds up to 10m/s (Mikkola et al, 2023; Schmidli et al, 2018), can we be sure that weather situations with a valley wind fall into the weak-wind category?**

Response: We agree that thermally-induced winds can reach large values, such as for valley winds, and even more so for persistent katabatic winds whose speed can surpass 20 m/s, e.g. in Antartica (e.g. Parish and Cassano, 2003). However, in our case where we focus on slope flows which are generally weak, for example the HCLIM3 mean wind speed for JJA at 12 UTC is 1 m/s and the 99th percentile is 2.9 m/s, so these indeed are predominantly weak winds. Nevertheless, we have cleaned the manuscript text to remove general statements about low wind speeds for thermal circulations and to emphasize that the term "weak winds" predominantly refers to synoptically weak winds, under which thermal circulations occur, or more specifically to slope flows.

**2) Horizontal resolution required to simulate slope flows (either anabatic or katabatic)**

**How can we be sure that the model at dx=3km is able to simulate anabatic or katabatic slope winds, given their small vertical extent and short-term variability? As Wagner et al (2014) suggest, at least 10 grid points across a valley are necessary to simulate the relevant thermally-induced flows, including slope winds. Furthermore, Schmidli et al (2018) and Goger and Dipankar (2024) show that at at least a horizontal grid spacing of 1km might be necessary to simulate the up-valley flow. However, slope flows might be present, but inaccurately simulated.**

Response: The mentioned studies focus on specific valleys equipped with meteorological observations and analyse the ability of a model to reproduce the observations, i.e. the realistic flow patterns. In this case, we agree that the horizontal resolution in relation to a specific valley geometry is crucial in reproducing the observed features. However, in our case the analysis is more general, not focusing on a specific valley or any other terrain feature, but rather taking into account all grid points in the same terrain altitude range, regardless of their position in the complex mountain environment (e.g. whether they are oriented toward the ocean or a plain, whether a valley exists or not, etc.). This approach removes a considerable amount of detailed flow information through bin averaging, including the distinction between mountain-plain, valley and slope flows. At the same time, it provides a more general and robust difference between the two models. In other words, in this study we are not concerned with the accuracy of each of the simulations with respect to observations but with the difference between the two models on climate scales, yielding a potential added value of the km-scale model. Following that, we do not claim that the results of the 3-km model are accurate, just conceptually more realistic than the 12 km model.

Therefore, the horizontal resolution is not an issue in this case, since at each resolution the model adapts the thermal circulations to the given topography, i.e. the flow is generally consistent with the topography at that resolution. A bigger issue could be the vertical resolution, given the low heights of katabatic jets, and also the typical model overdiffusion due to turbulence parameterization. It has been shown for idealised slope flows that a different formulation of turbulence parameterization can considerably improve the vertical structure of katabatic flow, even for 2-km horizontal resolution (Grisogono and Belušić, 2008). All this is now discussed in Section 3.4.

**3) The glacier wind**

**The authors often mention the katabatic glacier winds in their results. I am aware that glaciers are present in Southern Norway (Haualand et al, 2024). However, how well are the glaciers represented in the climate model at dx=3km? Are there ice surfaces with the according land-use categories (ice) and albedo (larger than 0.6) present?**

**Furthermore, even if the glaciers are present in the model, how well is a purely katabatically-driven flow represented in the model? To my current knowledge, even**

**models at the LES range (dx=50m or less, e.g., Goger et al, 2022) struggle to represent katabatic winds over glaciers (mostly due to too coarse vertical levels). Furthermore, Cuxart (2015) states that dx=5m is necessary to simulate stable boundary layers and katabatic flows.**

**Furthermore, at several occasions in the manuscript, the term "glacier wind" is used, while mostly katabatically-forced downslope winds are described, so it might make sense to stay with the term "katabatic winds".**

Response: For the resolution issue, we point to the response to comment 2. Given the discussion there, we argue that the horizontal resolution is not the key problem in our approach, since we are not trying to reproduce specific thermal flows in given valleys or other mountain features. The vertical resolution is most certainly an issue for stable boundary layer flows, which we now discuss in the paper. However, the main result is that unlike HCLIM12, HCLIM3 generates downslope wind in summer daytime conditions over snow-covered surfaces, which is conceptually consistent with the theory and observations. The accuracy of the structure of this downslope wind is not addressed here.

Regarding the term "glacier wind", we have referred to all grid points with snow cover in summer as „glaciers". There are glaciers in the model as land use categories, that are frequently co-located with the summer snow-covered surfaces, but we have not used that definition here. Given that this specific katabatic wind occurs in summer daytime, it has the characteristic of glacier winds in that it does not have the otherwise typical diurnal direction change between downslope and upslope wind. Furthermore, since this wind occurs in summer daytime, it is related to perpetual snow, and we assume that the difference between the effects of snow (summer snow) and ice (glacier) surface on the generation of downslope wind is small. So for lack of a better term, we use "glacier wind" for the perpetual-snow katabatic wind. This is now discussed in Section 3.4.

**Minor comments**

**line 45: "large-scale winds accelerated by terrain": What do you mean exactly? Downslope windstorms, such as foehn and bora?**

Response: We meant a more general range of flows accelerated by terrain, which includes downslope windstorms as the most prominent example, but also other types of flows, such as gap flows, acceleration due to mountain waves, etc. This is now stated in the manuscript text.

**Section 2.1, Model simulations: This section would benefit if a table with the different simulations and their configurations were added**

Response: The table has been added in Appendix A.

**lines70-75: How exactly did you categorize the weather stations into the categories (flat/coast/mountains)? If an extra Figure would be too much addition to the manuscript, you could also add it as a supplementary figure.**
Response: The underlying grouping of the stations comes from the categorization into mountainous, coastal and inland originally defined by the SMHI observational and weather forecasting departments. We have now included a map, Figure A1 in Appendix A, that shows the spatial distribution of the SMHI weather stations used in the analysis together with the orography from the 3 km HCLIM model.

**line158: " The presence of snow cover generally lowers the surface temperature compared to the equivalent conditions without snow cover and can cause downslope glacier wind even in daytime conditions." - This is the first occasion where you mention glacier winds, but over snow-covered surfaces (independent of whether there is a glacier below the snow cover) it would make sense to stay with the more general term "katabatic winds".**
Response: As argued in our response to comment 3, we prefer the term "glacier wind" because it better describes the characteristics of the wind in question, namely that it does not have the diurnal wind reversal (i.e. it is always downslope) and that the negative buoyancy, especially during summer daytime, is a result of similar surface characteristics as for glacier wind (snow or ice). However, if the referee still thinks that using "glacier wind" is not appropriate in this context, we can modify it to some other term that indicates the specific nature of the wind, such as "summer-snow katabatic wind", or similar.

**line180: "confirming the upslope anabatic nature of the summer daytime circulation" - the positive vertical velocities could also be associated with the development of a convective boundary layer in the valleys (e.g., Goger et al 2024, Their Fig...)**
Response: We agree that this could be the case in situations where the model resolution enables a proper reproduction of valley flows. In our more general case, the positive vertical velocities are predominantly related to upslope flows (Fig. A2 in Appendix A). Note also that the vertical velocities in our study are reported at just above the ground level along the slopes.

**line183: "This suggests that the downslope flow direction is the result of the katabatic glacier wind" - here is a second mention of the glacier wind, but it seems only to appear during winter when snow cover increases. Would the term katabatic wind be enough?**
Response: The text on line 182-183 in the original manuscript refers to the summer daytime conditions. To clarify this, in the revised manuscript we indicate that the discussed snow cover increase is relative to terrain height. Since this mention of "glacier wind" is related to summer daytime conditions, just like the other mentions, we refer to our response to comment 3.

**line255: " sub-km resolution is advantageous or even crucial (e.g. Wang et al., 2013)." - this is in agreement with the findings of Goger and Dipankar (2024), who conducted real-case simulations across the hectometric range of thermally-induced flows**
Response: We thank the referee for the reference; we now also cite Goger and Dipankar (2024).

**line261: Where do you see in your data the occurrence of "glacier fronts"?**
Response: We now clarify in the manuscript that the term "glacier wind" refers to katabatic wind in summer daytime situations, so these are not actually glacier fronts. However, we do see the signatures of frontal zones in the horizontal wind field and convergence (e.g. Figs. 8 and A2), vertical velocity and the cloud cover (Fig. 6).

**Figures 2 and 3: Please add the variable name to the units.**
Response: Done.

References:

Belušić Vozila, A., Belušić, D., Telišman Prtenjak, M. et al., 2024: Evaluation of the near-surface wind field over the Adriatic region: local wind characteristics in the convection-permitting model ensemble. Clim Dyn 62, 4617–4634. https://doi.org/10.1007/s00382-023-06703-z

Chen, T.-C., Collet, F., & Di Luca, A., 2024: Evaluation of ERA5 precipitation and 10-m wind speed associated with extratropical cyclones using station data over North America. International Journal of Climatology, 44(3), 729–747. https://doi.org/10.1002/joc.8339

Fan, W., Y. Liu, A. Chappell, L. Dong, R. Xu, M. Ekström, T. Fu, and Z. Zeng, 2021: Evaluation of Global Reanalysis Land Surface Wind Speed Trends to Support Wind Energy Development Using In Situ Observations. J. Appl. Meteor. Climatol., 60, 33–50, https://doi.org/10.1175/JAMC-D-20-0037.1

Gandoin, R. and Garza, J., 2024: Underestimation of strong wind speeds offshore in ERA5: evidence, discussion and correction, Wind Energ. Sci., 9, 1727–1745, https://doi.org/10.5194/wes-9-1727-2024

Grisogono, B., Belušić, D., 2008: Improving mixing length-scale for stable boundary layers. Q. J. R. Meteorol. Soc., 134, 2185 – 2192, doi: 10.1002/qj.347
Molina MO, Gutiérrez C, Sánchez E., 2021: Comparison of ERA5 surface wind speed climatologies over Europe with observations from the HadISD dataset. Int J Climatol., 41: 4864–4878. https://doi.org/10.1002/joc.7103

Molina MO, Gutiérrez C, Sánchez E., 2021: Comparison of ERA5 surface wind speed climatologies over Europe with observations from the HadISD dataset. *Int J Climatol*., 41: 4864–4878. https://doi.org/10.1002/joc.7103

Parish, T. R., and J. J. Cassano, 2003: The Role of Katabatic Winds on the Antarctic Surface Wind Regime. Mon. Wea. Rev., 131, 317–333, doi: 10.1175/1520-0493(2003)131<0317:TROKWO>2.0.CO;2

Potisomporn, P., Adcock, T.A.A., Vogel, C.R., 2023: Evaluating ERA5 reanalysis predictions of low wind speed events around the UK. Energy Reports, 10, 4781–4790, https://doi.org/10.1016/j.egyr.2023.11.035

Van den Broeke, M. R., 1997: Structure and diurnal variation of the atmospheric boundary layer over a mid-latitude glacier in summer. Boundary-Layer Meteorology 83, 183–205, doi: 10.1023/A:1000268825998

Zardi, D., Whiteman, C.D., 2013: Diurnal Mountain Wind Systems. In: Chow, F., De Wekker, S., Snyder, B. (eds) Mountain Weather Research and Forecasting. Springer Atmospheric Sciences. Springer, Dordrecht, doi: 10.1007/978-94-007-4098-3_2